# Oxygen Glucose Deprivation-Induced Lactylation of H3K9 Contributes to M1 Polarization and Inflammation of Microglia Through TNF Pathway

**DOI:** 10.3390/biomedicines12102371

**Published:** 2024-10-17

**Authors:** Lu He, Rui Yin, Weijian Hang, Jinli Han, Juan Chen, Bin Wen, Ling Chen

**Affiliations:** 1Division of Neonatology, Department of Pediatrics, Tongji Hospital, Tongji Medical College, Huazhong University of Science and Technology, Wuhan 430022, China; d202282184@hust.edu.cn; 2Department of Biochemistry and Molecular Biology, School of Basic Medicine, Tongji Medical College, Huazhong University of Science and Technology, Wuhan 430022, China; yinrui75426@163.com (R.Y.); chenjuanlinda69@163.com (J.C.); 3Division of Cardiology, Department of Internal Medicine, Tongji Hospital, Tongji Medical College, Huazhong University of Science and Technology, Wuhan 430022, China; hangwjcardio@tjh.tjmu.edu.cn; 4Department of Pediatrics, Shanxi Bethune Hospital, Shanxi Academy of Medical Sciences, Tongji Shanxi Hospital, Third Hospital of Shanxi Medical University, Taiyuan 030032, China; a1181093873@163.com; 5Department of Clinical Laboratory, Shanxi Bethune Hospital, Shanxi Academy of Medical Sciences, Tongji Shanxi Hospital, Third Hospital of Shanxi Medical University, Taiyuan 030032, China

**Keywords:** microglia, polarization, histone H3K9 lactylation, neonatal hypoxic–ischemic encephalopathy, neuroinflammation

## Abstract

Background: Hypoxia-induced M1 polarization of microglia and resultant inflammation take part in the damage caused by hypoxic-ischemic encephalopathy (HIE). Histone lactylation, a novel epigenetic modification where lactate is added to lysine residues, may play a role in HIE pathogenesis. This study investigates the role of histone lactylation in hypoxia-induced M1 microglial polarization and inflammation, aiming to provide insights for HIE treatment. Methods: In this study, we assessed the effects of hypoxia on microglial polarization using both an HIE animal model and an oxygen–glucose deprivation cell model. Histone lactylation at various lysine residues was detected by Western blotting. Microglial polarization and inflammatory cytokines were analyzed by immunofluorescence, qPCR, and Western blotting. RNA sequencing, ChIP-qPCR, and siRNA were used to elucidate mechanisms of H3K9 lactylation. Results: H3K9 lactylation increased due to cytoplasmic lactate during M1 polarization. Inhibiting P300 or reducing lactate dehydrogenase A expression decreased H3K9 lactylation, suppressing M1 polarization. Transcriptomic analysis indicated that H3K9 lactylation regulated M1 polarization via the TNF signaling pathway. ChIP-qPCR confirmed H3K9 lactylation enrichment at the TNFα locus, promoting OGD-induced M1 polarization and inflammation. Conclusions: H3K9 lactylation promotes M1 polarization and inflammation via the TNF pathway, identifying it as a potential therapeutic target for neonatal HIE.

## 1. Introduction

Hypoxic–ischemic encephalopathy (HIE) is a neurological condition caused by oxygen deprivation to the infant brain [1]. Neuroinflammation significantly contributes to the pathophysiology of perinatal brain injury [2,3,4]. Under oxygen deprivation, microglia, the central nervous system’s immune cells, become rapidly activated and polarized, releasing inflammatory cytokines that induce neuroinflammation and brain injury. Microglia can polarize into M1 and M2 subtypes, with M1 microglia releasing cytokines such as tumor necrosis factor-α (TNFα), interleukin-1β (IL-1β), and interleukin-6 (IL-6) [5]. These cytokines induce neuroinflammation and promote neuronal damage, playing a crucial role in HIE [6,7,8]. M2 microglia can promote tissue repair and regeneration [9]. Thus, understanding the regulatory mechanisms of M1 polarization in microglia is crucial for preventing their pathogenic effects in neonatal HIE.

Epigenetic modifications regulate biological processes by adding or hydrolyzing functional groups in histones and DNA, influencing gene replication, transcription, and translation [10,11,12]. Histones, basic structural proteins of eukaryotic chromosomes, undergo post-translational modifications like methylation and acetylation, affecting their affinity for DNA, altering chromatin structure, and regulating gene expression [13,14]. Recent studies have identified lactylation [15,16] as a novel epigenetic modification, where lactate groups are added to lysine residues on histones.

Research indicates that histone lactylation can regulate neuroinflammatory phenotypes. Mao Haiping et al. found that histone lactylation promotes inflammation by activating the NF-κB pathway [17], while Zhao Yingming et al. discovered that histone lactylation increases the M2 marker Arg1 in macrophages [15]. During HIE, hypoxia induces significant lactate accumulation in microglia [18], but it is unclear whether lactate can act as a substrate to promote histone lactylation, thereby regulating microglial polarization.

In this study, we demonstrate that oxygen–glucose deprivation (OGD) induces increased lactate production in microglia. Mechanistically, we find that H3K9 lactylation regulates the TNF signaling pathway in microglia, promoting M1 polarization and the associated inflammatory response. Our findings provide new insights into mitigating neuroinflammation through histone epigenetic modifications following OGD.

## 2. Materials and Methods

### 2.1. Neonatal Hypoxic–Ischemic Encephalopathy Rat Model

The Rice–Vannucci rat model is extensively used to study neonatal HIE [19]. Sprague–Dawley (SD) neonatal rats were sourced from Shulaibao Biotechnology Co., Ltd. (Wuhan, China). P7 pups were anesthetized with 2% isoflurane, followed by exposure and double ligation of the left common carotid artery (CCA) using 5.0 silk sutures. After wound suturing, the pups recovered for 1 h before being exposed to an environment of 8% O_2_ and 92% N_2_ for 2 h at 37 °C. Subsequently, they were returned to their dam for recovery. In the sham group, the CCA was exposed without ligation, and no hypoxic treatment was applied. There were 20 rats in sham group or HIE model group. All procedures were approved by the Experimental Animal Ethics Committee of Tongji Medical College, Huazhong University of Science and Technology. The registration number of the ethical approval is [2023] IACUC Number:3908.

### 2.2. Cell Culture and Oxygen Glucose Deprivation Cell Model

BV2 microglial cells were obtained from Wuhan Servicebio Technology Co., Ltd. The cells were cultured in DMEM (Gibco, 11995065, New York, NY, USA) with 10% fetal bovine serum (Yeasen, 40131ES76, Shanghai, China) for two days. When they reached around 70% confluence, oxygen glucose deprivation (OGD) induction began by replacing the old medium and washing the cells three times with PBS before adding different media. The normal group received complete medium with 10% fetal bovine serum and was incubated at 37 °C with 5% CO_2_. In contrast, the OGD model group was exposed to low-glucose DMEM (Gibco, 6123033, New York, NY, USA) with 4% fetal bovine serum in a hypoxic chamber set to 37 °C, 5% CO_2_, 4% O_2_, and 91% N_2_. Cells were collected at various time points for subsequent experiments.

### 2.3. TTC Staining

Twenty-four hours post-hypoxia–ischemia, TTC staining was used to measure the cerebral infarction area. Rats were anesthetized with 2% isoflurane and decapitated. The brains were quickly dissected on ice and frozen at −80 °C for 5 min, then sliced according to mold size. The slices were then placed in TTC staining solution and incubated at 37 °C in the dark for 30 min, flipping every 5 min. After staining, the slices were washed twice with PBS and fixed with 4% paraformaldehyde for observation and photography. The infarct area was analyzed using software ImageJ 1.54f and calculated as: percentage of infarct area (%) = (infarct area/total brain tissue area) × 100%.

### 2.4. Cell Counting Kit-8 Viability Assay

BV2 cells were cultured for 24 h after being seeded at 5000 cells/well in 96-well plates. The following day, the medium was replaced. The total treatment duration was 24 h. Cells from different experimental groups were exposed to OGD treatment at the 1st, 13th, and 19th hours (as detailed in Section 2.2), alongside a control group maintained under standard culture conditions. After the treatment, 100 μL of Cell Counting Kit-8 (CCK8) working solution (Abbkine, ATWD2409, Wuhan, China) was added to each well. After 30 min of incubation, the absorbance at 450 nm was measured using a microplate reader. Cell viability was determined by normalizing the absorbance of each group with that of the blank group. All data were normalized using the mean of the 0H group.

### 2.5. Lactate Measurement

Lactate levels in tissues and cell pellets were measured using an L-lactate assay kit (Cat# A019-2-1, Nanjing Jiancheng Bioengineering Institute, Nanjing, China) according to the manufacturer’s instructions. The cells (as detailed in Section 2.2 and Section 2.8) were harvested and transferred to centrifuge tubes, then washed with pre-cooled PBS. The supernatant was removed after centrifugation, and 1 mL of lactate assay buffer was added. Cells were sonicated by ultrasound in an ice bath for 5 min (200 W, 3 s, 7 s intervals, 30 times). After centrifugation at 12,000× *g* for 5 min at 4 °C, the supernatant was transferred to a new EP tube for measurement. Ten milligrams of fresh hippocampal tissue (as detailed in Section 2.1) were ground in a high-speed tissue grinder (KZ-11, Servicebio, Wuhan, China) at 120 Hz for 2 min. The tissue was mixed with 1 mL of lactate assay buffer, homogenized in an ice bath, and centrifuged at 12,000× *g* for 5 min at 4 °C. Then, 2 μL of the supernatant was used to determine the protein concentration using the BCA kit (20201ES86, Yeasen). The rest of the supernatant was collected for measuring lactate levels using the L-lactate assay kit. The collected supernatant, distilled water, and 3 mM standard substance (3 mM L-lactate) were added into separate EP tubes. Then, the enzyme working solution and the chromogenic agent were added, mixed, and reacted in a water bath at 37 °C for 10 min. After reaction, the termination solution was added to each EP tube and thoroughly mixed. Finally, the absorbance was measured using a microplate reader at a wavelength of 530 nm. The content of lactate was calculated from the absorbance of the microplate reader compared with the protein concentration. The computational formula was as follows: (absorbance of sample − absorbance of distilled water)/(absorbance of standard substance − absorbance of distilled water) × concentration of the standard/protein concentration of the sample. All concentrations of samples were normalized using the mean value of the control group.

### 2.6. Immunofluorescence Staining

Twenty-four hours after modeling (as detailed in Section 2.1), rats were anesthetized and perfused transcardially with ice-cold 4% paraformaldehyde. The brains were fixed in the same solution for 24 h, then placed in 30% sucrose for dehydration over 3 days. Coronal sections, 15 μm thick, were cut at −20 °C using a cryostat (FS800, RWD, Shenzhen, China), mounted on slides, and subjected to immunofluorescence staining. Sections were blocked with 10% goat serum and incubated with primary antibodies (anti-Iba1, Servicebio, GB12105-10; ant-LDHA, Proteintech, 19987-1-AP, Wuhan, China; anti-H3K9 lactylation, PTM BIO, PTM-1419RM, Hangzhou, China; anti-iNOS, Proteintech, 18985-1-AP), followed by secondary antibodies (Alexa Fluor 488 and Cy3 conjugates, Servicebio, GB25303; GB21301; GB25301; GB21303), for 1 h. For cell samples, cells were grown on confocal dishes (Biosharp, BS-15-GJM, Beijing, China), fixed with 4% paraformaldehyde, permeabilized with 0.1% Triton X-100, and blocked with 5% BSA. Cells were then incubated with primary antibodies (anti-H3K9 lactylation, PTM BIO, PTM-1419RM; anti-iNOS, Proteintech, 18985-1-AP) and secondary antibodies (Alexa Fluor 488 and Cy3 conjugates, GB25303 and GB21303). Confocal microscopy (Nikon, AX/AX R with NSPARC, Japan, China) was used for imaging with average fluorescence intensity measured by ImageJ 1.54f software, and iNOS-positive cells were counted manually. The through-focus images for z-stacks were generated from confocal microscopy (Nikon, AX/AX R with NSPARC). Pearson’s coefficient of colocalization analysis was performed using a plugin named JACoP [20] in ImageJ. The total number of cells shown by DAPI was used to calculate the percentage of cells expressing positive proteins. Anti-LDHA, anti-H3K9 lactylation, and anti-iNOS were prepared from rabbit. Anti-Iba1 was prepared from mouse. The secondary antibody conjugated to Alexa Fluor 488 (GB25303) and Cy3 (GB21303) was prepared from goat, which was anti-rabbit. The secondary antibody conjugated to Alexa Fluor 488 (GB25301) and Cy3 (GB25301) was prepared from goat, which was anti-mouse.

### 2.7. Western Blotting

Protein extraction was performed using RIPA lysis buffer containing deacetylase inhibitors (MCE, HY-K0030, Shanghai, China) and phenylmethylsulfonyl fluoride (PMSF). The sample was lysed for 15 min on the ice, then centrifuged at 16,000× *g* for 15 min at 4 °C. We measured protein concentration of the supernatant via the BCA kit (20201ES86, Yeasen) and adjusted the protein concentration to be consistent by adding RIPA lysis buffer. Finally, loading buffer was added, and a metal bath at 100 °C was used for 10 min. Equal amounts of protein were separated by 10% SDS-PAGE and transferred to nitrocellulose membranes. Membranes were blocked with 5% non-fat milk and incubated with primary antibodies (anti-Pan Kla, PTM BIO, PTM-1401; anti-Actin, Abclonal, AC026, Wuhan, China; anti-H3, Abclonal, A2352; anti-H4, Abclonal, A23000; anti-iNOS, Proteintech, 18985-1-AP; anti-CD206, Abclonal, A8301; anti-H3K9 lactylation, Abclonal, A18827; anti-H3K18la, Abclonal, A21214; anti-H3K27la, Abclonal, A18825; anti-H3K56la, Abclonal, A18826; anti-H4K16la, Abclonal, A18828; anti-H4K8la, Abclonal, A18830) overnight, followed by TBST washing. Membranes were then incubated with secondary antibodies (Abclonal, AS014) and visualized using ECL in a dark room. Protein expression levels were evaluated by densitometry using ImageJ 1.54f software and normalized accordingly. The expression level of histone 3 modifications was relatively quantified by the expression level of histone 3, and the expression level of histone 4 modifications was relatively quantified by the expression level of histone 4. The expression of other proteins was relatively quantified using the expression of Actin. All data were normalized using the control group. All primary antibodies were prepared from rabbit. The secondary antibody conjugated was prepared from goat, which was anti-rabbit.

### 2.8. siRNA Transfection

siRNA or negative control siRNA was transfected using non-liposome PEI transfection reagent (40806ES, Yeasen) following the standard protocols. BV2 cells were transfected in 6-well plates at 80% confluence with 75 nM control siRNA (NC) or LDHA-targeted siRNA (siLDHA) (sense, 5′-CAGCAAAGACUACUGUGUATT-3′; antisense, 5′-UACACAGUAGUCUUUGCUGTT-3′), TNFα-targeted siRNA (siTNFα) (sense, 5′-GCAUGGAUCUCAAAGACAATT-3′; antisense, 5′-UUGUCUUUGAGAUCCAUGCTT-3′). BV2 cells were seeded 14–18 h before transfection to ensure the required 70–80% confluence at the time of collection. After 24 h of transfection, the cells showed normal morphology and no significant cell death was observed, and then cells were treated under control conditions (DMEM medium, 10% FBS) or OGD conditions.

### 2.9. Real-Time Quantitative PCR

Collected cells were lysed with Trizol reagent. Half the trizal volume of chloroform was added, and then we waited for 5 min. The mixture was centrifuged at 12,000× *g* for 10 min at 4 °C to reach stratification. Then supernatant was mixed with an equal volume of isopropyl alcohol. After centrifugation at 12,000× *g* for 10 min at 4 °C, the sediment was washed twice with 75% ethanol, and mRNA was dissolved in ddH_2_O. cDNA was synthesized using a commercial reverse transcription kit (BL699A, Biosharp). Real-time quantitative PCR (qPCR) analysis was performed using a commercial qPCR kit (Abclonal, RK20404) with a qPCR detection system (Bio-Rad, 1855201, Shanghai, China). The following primers were used: *Actin*, ACCTTCTACAATGAGCTGCG/CTGGATGGCTACGTACATGG; *Inos*, GTTCTCAGCCCAACAATACAAGA/GTGGACGGGTCGATGTCAC; *Cd206*, CTCTGTTCAGCTATTGGACGC/CGGAATTTCTGGGATTCAGCTTC; *Ldha*, TGTCTCCAGCAAAGACTACTGT/GACTGTACTTGACAATGTTGGGA; *Mcp1*, TTAAAAACCTGGATCGGAACCAA/GCATTAGCTTCAGATTTACGGGT; *Il-6*, CCAAGAGGTGAGTGCTTCCC/CTGTTGTTCAGACTCTCTCCCT; *Tnfα*, CCCTCACACTCAGATCATCTTCT/GCTACGACGTGGGCTACAG; *Gm-csf*, GGCCTTGGAAGCATGTAGAGG/GGAGAACTCGTTAGAGACGACTT; *Il-18*, and GACTCTTGCGTCAACTTCAAGG/CAGGCTGTCTTTTGTCAACGA. mRNA expression levels were relatively quantified by the *Actin* expression levels. All data were normalized using the control group.

### 2.10. RNA Library Preparation and Sequencing

Samples were placed in Trizol and stored at −80 °C. Subsequent sequencing techniques and analysis were supported by Novogene. According to the manufacturer’s instructions, RNA integrity was assessed using the RNA Nano 6000 Assay Kit of the Bioanalyzer 2100 system (Agilent Technologies, Santa Clara, CA, USA). Clean data were obtained by removing reads containing adapters, reads containing N base, and low-quality reads from raw data. FeatureCounts v1.5.0-p3 was used to count the read numbers mapped to each gene. Then, the FPKM of each gene was calculated based on the length of the gene and the read count mapped to this gene. The R Bioconductor package DESeq2 was utilized to screen out differentially expressed genes (DEGs). The *p* value < 0.01 and |log2fc| > 0.5 were set as the cut-off criteria for identifying DEGs. Gene Ontology (GO) terms and Kyoto Encyclopedia of Genes and Genomes (KEGG) pathways were identified using the KOBAS 2.0 server to sort out functional categories of DEGs.

### 2.11. Chromatin Immunoprecipitation

Chromatin immunoprecipitation (ChIP) was conducted using a commercial ChIP kit (Abclonal, RK20258). Cells (as detailed in Section 2.2) were collected from the culture dish using a cell scraper with PBS, and the supernatant was removed after centrifugation. The next step was stabilizing interactions between proteins and DNA within chromatin using formaldehyde crosslinking. Subsequent steps included extracting cell nuclei from samples, followed by sonication to shear chromatin into appropriately sized DNA fragments. Specific antibodies (anti-H3K9 lactylation, PTM BIO, PTM-1419RM) were then used for immunoprecipitation, binding to target proteins. Washing steps removed non-specifically bound proteins and other contaminants. Crosslinks were reversed, and enzyme digestion (Abclonal, RK20258) was used to remove protein crosslinks, allowing the DNA fragments to dissociate. Finally, DNA was purified using a commercial DNA extraction kit (Abclonal, RK30100) and analyzed by qPCR to quantify or qualitatively assess the binding of *Tnfα*.

### 2.12. Statistical Analysis

Statistical analysis was conducted using GraphPad Prism 8.0.2 software. Data were compared using the unpaired Student’s *t*-test for two groups, while one-way ANOVA with Tukey’s post hoc test was used to compare data between multiple groups. Statistical significance was set at *p* < 0.05.

## 3. Results

### 3.1. Hypoxia Promotes M1 Polarization of Microglia

To determine whether microglia undergo polarization and to identify the type of microglia polarization in HIE, we established an HIE model in 7-day-old SD rat pups, with sham-operated pups as the control group. Compared to the sham group, TTC staining demonstrated a pronounced infarct on the left hemisphere of the brain in the HIE group (Figure 1A). Immunofluorescence detection showed a significant increase in the M1 microglia marker iNOS in the HIE group compared to the sham group (Figure 1B). Western blotting assessments confirmed a significant increase in iNOS and a decrease in CD206, the M2 microglia marker, in HIE versus sham brain tissues (Figure 1C). In vitro, immunofluorescence analysis revealed a substantial elevation in iNOS (+) microglia in the OGD model (Figure 1D). qPCR and Western blotting analysis also indicated a significant increase in iNOS protein and mRNA expression levels and a significant decrease in CD206 protein and mRNA expression levels after OGD (Figure 1E,F).

Collectively, these findings indicated that neonatal hypoxia leads to brain tissue infarction and fosters M1 polarization of microglia.

### 3.2. Lactate Accumulates During Oxygen–Glucose Deprivation in Microglia

During OGD, cells underwent metabolic reprogramming to initiate anaerobic glycolysis [21]. Brain tissues from HIE neonatal rats exhibited significantly elevated lactate levels compared to their sham-operated counterparts, reflecting enhanced anaerobic glycolysis (Figure 2A). In BV2 microglial cells subjected to OGD, lactate accumulation increased progressively over time (Figure 2B). Concurrently, CCK8 demonstrated a decline in cell survival with extended OGD exposure (Figure 2C). Notably, after 24 h of OGD, *Ldha* mRNA expression in BV2 cells was significantly elevated compared with controls (Figure 2D). LDHA catalyzes the terminal step in lactate production during anaerobic glycolysis [22,23]. Western blotting analysis showed a significant upsurge in LDHA protein levels (Figure 2E), and immunofluorescence indicated a marked increase in LDHA in Iba1- positive microglia within the HIE animal model (Figure 2F). In summary, these findings from both the animal and cellular models suggest an upregulation of LDHA, facilitating increased lactate production by microglia.

### 3.3. Increased H3K9 Lactylation Following Oxygen Glucose Deprivation in Microglia

Total protein lactylation levels in BV2 cell gradually increased during OGD (Figure 3A). Analysis of histone lactylation sites showed a significant elevation in H3K9 lactylation (Figure 3B). Immunofluorescence staining of microglia (Iba1-positive) in both the BV2 OGD model and the SD rat HIE model demonstrated a marked increase in H3K9 lactylation (Figure 3C,D). However, the evaluation of total protein acetylation and H3K9 acetylation levels in BV2 cells during OGD revealed no significant changes (Figure 3E).

### 3.4. H3K9 Lactylation Activates the TNF Signaling Pathway to Regulate M1 Polarization and Inflammation in Microglia

Previous studies have demonstrated that the histone acetyltransferase P300 mediates histone lactylation by transferring lactyl groups to lysine residues on histones [24,25]. At the beginning of the OGD, we added 50 μM P300 inhibitor, A-485 (MCE, HY-107455), for 24 h to significantly suppress OGD-induced H3K9 lactylation (Figure 4A). Immunofluorescence and Western blotting analyses showed that P300 inhibition reduced the expression of the cell marker iNOS in OGD-induced M1-type microglia (Figure 4B,C). These findings suggest that reducing H3K9 lactylation levels may attenuate M1 polarization in microglia.

To elucidate the molecular mechanisms involved, we performed transcriptomic analysis (Figure 5A–E). We compared the differentially expressed genes (DEGs) that were upregulated in the OGD group versus the control group and downregulated in the OGD-A485 group versus the OGD group (*p* < 0.01, |log2fc| > 0.5), identifying 504 genes shown in a Venn diagram (Figure 5B). Among the 504 DEGs, we found that OGD-induced M1 microglia-specific genes were all suppressed by P300 inhibitor treatment (Figure 5C). KEGG enrichment analysis of these 503 genes revealed significant enrichment in the TNF signaling pathway (Figure 5D). GO enrichment analysis indicated significant enrichment in biological processes regulating inflammation (Figure 5E). The TNF signaling pathway, a classical inflammatory pathway, is known for its role in activating M1 macrophage polarization [26]. We further examined the expression of *Tnfα*, a key molecule in this pathway, using Western blotting and qPCR. Both mRNA and protein levels of TNFα significantly increased after OGD and decreased significantly following treatment with the P300 inhibitor (Figure 5F,G). Additionally, qPCR results showed that several pro-inflammatory genes, including *Gm-csf*, *Mcp1*, and *Il-6*, were upregulated after OGD and downregulated by the P300 inhibitor (Figure 5G). These findings suggest that H3K9 lactylation may promote M1 microglial polarization and inflammation through the TNF signaling pathway.

To further elucidate the role of the TNF signaling pathway in H3K9 lactylation-mediated M1 microglial polarization and inflammation, we performed ChIP-qPCR and found significant enrichment of H3K9 lactylation at the *Tnfα* gene locus after OGD (Figure 5H). Using siRNA to knock down TNFα significantly inhibited OGD-mediated M1 microglial polarization and the increase in inflammatory factors (Figure 5I,J). In conclusion, after OGD, H3K9 lactylation promoted M1 microglial polarization and inflammation by activating the transcription of *Tnfα*, thereby enhancing the TNF signaling pathway.

### 3.5. Inhibition of Lactate Production Alleviates Oxygen Glucose Deprivation-Mediated M1 Microglial Polarization and Inflammation

Histone lactylation uses lactate, a product of anaerobic glycolysis, as a substrate [24,27]. Since P300 modifies both lactylation and acetylation, we used siRNA to knock down LDHA, significantly reducing lactate secretion in BV2 cells after OGD (Figure 6A). LDHA knockdown also markedly decreased total protein lactylation and H3K9 lactylation levels induced by OGD (Figure 6B). Additionally, the expression of M1 microglial marker and various inflammatory factors significantly decreased following LDHA knockdown (Figure 6C–E). The increase in TNFα expression mediated by OGD was also inhibited by the knockdown of LDHA (Figure 6F).

In summary, reducing the production of lactate, the substrate for H3K9 lactylation, can alleviate OGD-induced M1 microglial activation and inflammation.

## 4. Discussion

Microglia-mediated neuroinflammation is a major contributor to secondary neuronal injury in HIE [28,29,30]. Under oxygen–glucose deprivation, microglia undergo M1 polarization and secrete numerous inflammatory factors, exacerbating brain injury [8,31]. Additionally, metabolic reprogramming in HIE significantly increases lactate production, which further promotes inflammation [32,33]. In this study, we investigated the metabolic reprogramming of microglia and its impact on the inflammation following OGD.

In the context of HIE, we observed a phenotypic shift towards M1 polarization in microglia within the rat brain. The prevalence of M1-type microglia is often associated with neuroinflammation, suggesting that HIE may exacerbate brain tissue damage by promoting M1 activation of microglia. Subsequent research revealed a significant accumulation of lactate, a byproduct of anaerobic glycolysis, in microglia following OG. This indicates an adaptive response to energy demands in a hypoxic environment through enhanced lactate production. Growing evidence supports the role of metabolic reprogramming in regulating innate inflammation. In classically activated M1 macrophages and dendritic cells, there is a metabolic shift towards glycolysis, leading to increased glucose uptake and lactate production, as well as the generation of nitric oxide and citrulline [34,35]. Elevated glycolytic activity and lactate levels contribute to pro-inflammation responses during lipopolysaccharide-induced polarization of BV2 cells [33].

The TNF signaling pathway, a classic inflammatory pathway, plays a crucial role in M1 polarization of microglia and has garnered widespread attention [36,37]. Wu Zhongliang et al. demonstrated that the TNF signaling pathway mediates microglial activation in acute paradoxical sleep deprivation in mice [36]. TNFα, a key ligand in the TNF signaling pathway, can alleviate microglia-mediated neuroinflammation by inhibiting its production [38]. However, it remains unclear whether metabolic reprogramming of microglia following OGD regulates polarization and inflammation through the TNF signaling pathway or TNFα, necessitating further investigation into the underlying molecular mechanisms.

Our research revealed that OGD leads to lactylation of histone H3K9 in microglia, possibly mediated by P300. Inhibiting P300 significantly reduced H3K9 lactylation levels and decreased M1 polarization of microglia post-OGD. These data suggest that histone H3K9 lactylation may play a role in regulating M1 polarization of microglia. KEGG enrichment analysis of the transcriptome highlighted the TNF signaling pathway, while GO enrichment analysis indicated a significant enrichment in biological processes regulating inflammation. Given that the TNF signaling pathway is a classic regulatory pathway for M1 microglial polarization, we proposed that H3K9 lactylation activates M1 microglial polarization and inflammation through this pathway. ChIP-qPCR and siRNA knockdown experiments further demonstrated that H3K9 lactylation promotes M1 polarization and inflammation in microglia by activating TNFα transcription, thereby mediating the TNFα signaling pathway.

TNFα, a key ligand in the TNF signaling pathway, activates downstream signaling by binding to TNFR1 or TNFR2 [39,40]. However, the biological effects downstream of TNFR1 and TNFR2 differ. TNFR1 primarily regulates apoptosis, necrosis, and inflammation, while TNFR2 is mainly involved in cell repair, regeneration, and immune regulation. Numerous studies have shown that TNFR1 and TNFR2 exhibit antagonistic effects during inflammation [41,42]. Our transcriptomic data indicated that differentially expressed genes were mainly enriched in molecules downstream of the TNFα and TNFR1 pathways (including *Ptgs2*, *Ccl2*, *Edn1*, *Lif*, etc.). According to certain studies, TNFα exists in two forms: transmembrane TNFα (tmTNFα, 25 kDa) and soluble TNFα (sTNFα, 17 kDa) [43]. These isoforms have different affinities for TNFR1 and TNFR2, with sTNFα activating inflammatory factors like IL-1β and IL-6 via TNFR1 [44]. However, it remains unclear why TNFα activates TNFR1 rather than TNFR2 in our HIE and OGD models, necessitating further investigation.

Research on H3K9 lactylation in regulating inflammatory factors is limited, but studies on H3K9 acetylation in macrophages have garnered attention [45,46,47]. Shihab Kochumon et al. found that H3K9 acetylation can increase IL-6 transcription levels in monocytes [46], and Chuanlong Wang et al. discovered that H3K9 acetylation can sustain IL-1β expression [47]. H3K9 acetylation and lactylation, being different modifications at the same site, may regulate the transcription of the same genes. Therefore, we examined the acetylation levels in our experimental models, but observed no significant changes. We hypothesize that increased lactylation in microglia under OGD is due to elevated lactate levels. During OGD, increased LDHA expression leads to extensive conversion of pyruvate to lactate, limiting the availability of acetyl-CoA necessary for acetylation. Consequently, no increase in acetylation was observed.

Finally, by knocking down LDHA, a key enzyme in lactate production, we obtained similar results as with P300 inhibitors, ruling out the possibility that P300 inhibitors regulate polarization and inflammation through acetylation. This further supports our conclusion.

In this study, we examined four lactylation sites on histone 3 and two lactylation sites on histone 4, finding that H3K9 lactylation was the most significantly upregulated. However, whether lactylation at other unexamined histone sites also plays important roles in macrophage polarization and inflammation requires further investigation.

In summary, our results emphasize the impact of histone lactylation on microglial M1 polarization and inflammation under oxygen–glucose deprivation. Our study provides new mechanistic insights into neuroinflammation induced by oxygen–glucose deprivation and proposes potential therapeutic strategies to mitigate inflammation-related damage in HIE and promote neuroprotection. Future research should further explore the clinical application of these mechanisms.

## Figures and Tables

**Figure 1 biomedicines-12-02371-f001:**
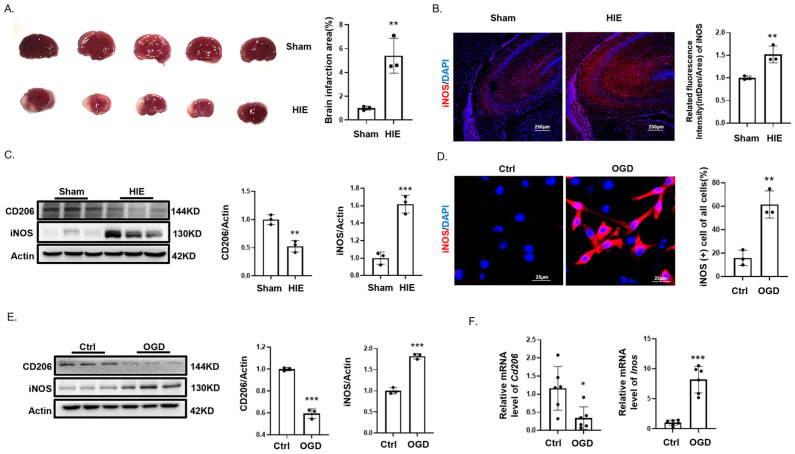
Hypoxia-Induced M1 microglial polarization. (**A**) TTC staining highlights an extensive infarct area in the left hemisphere of the brain in the hypoxic–ischemic encephalopathy (HIE) model (*n* = 3). (**B**) Immunofluorescence of iNOS (+) microglial highlights increased in the infarct zone compared to sham. Scale bar = 250 μm (*n* = 3). (**C**) Western blotting of CD206 and iNOS shows differential expression in HIE versus sham brain tissues (*n* = 3). (**D**) Immunofluorescence shows a significant increase in M1 polarization of BV2 cells after 24 h of oxygen glucose deprivation (OGD) (*n* = 100). Scale bar = 25 μm. (**E**) Western blotting shows changes in CD206 and iNOS levels in BV2 cells after 24 h OGD (*n* = 3). (**F**) qPCR quantifies changes in *Cd206* and *Inos* mRNA levels in BV2 post-OGD (*n* = 6). Significance; * *p* < 0.05; ** *p* < 0.01; *** *p* < 0.001.

**Figure 2 biomedicines-12-02371-f002:**
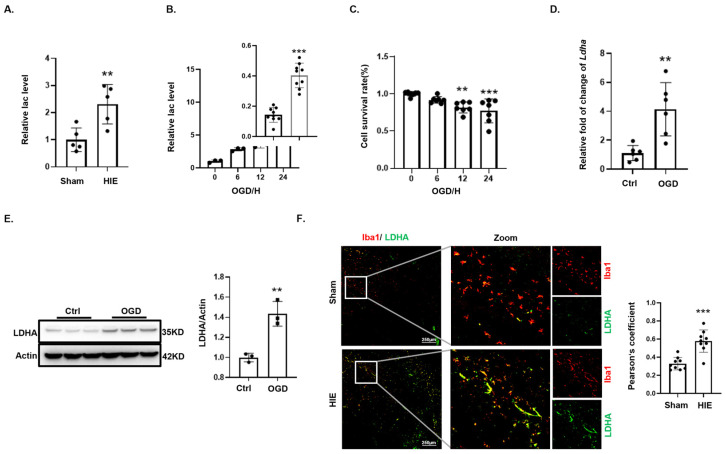
Effects of glucose deprivation on microglial anaerobic metabolism. (**A**) Lactate assay comparison between brain tissues from subjected to HIE and sham controls (*n* = 3). (**B**) Lactate quantification in BV2 cell over time during OGD, normalized relative to cell counts (*n* = 3). (**C**) Cell viability changes during OGD assessed by CCK8 assays (*n* = 6). (**D**) *Ldha* mRNA levels in BV2 cells post-OGD measured via qPCR (*n* = 6). (**E**) LDHA protein levels in BV2 cells post-OGD analyzed by Western blotting (*n* = 3). (**F**) LDHA expression in microglia (Iba1-positive) from sham and HIE groups shown by immunofluorescence, with Iba1 in red and LDHA in green. Scale bar = 100 μm (*n* = 9). Significance: ** *p* < 0.01; *** *p* < 0.001.

**Figure 3 biomedicines-12-02371-f003:**
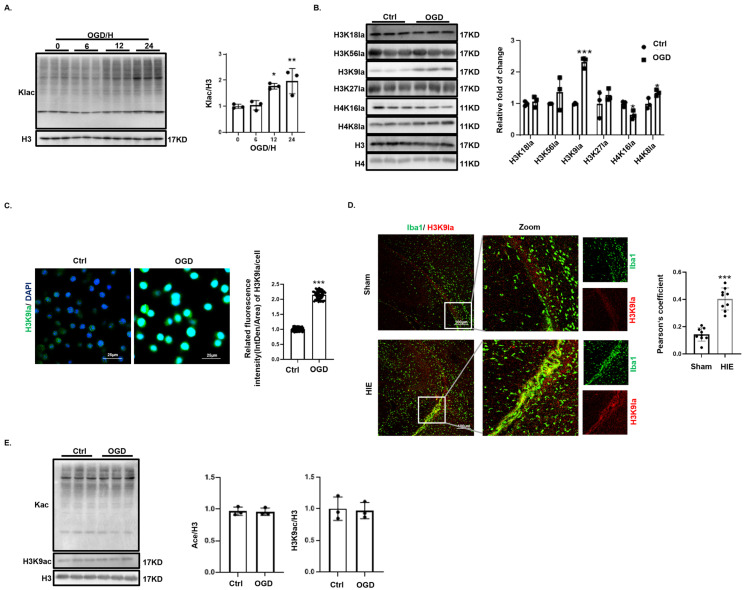
Increased H3K9 lactylation following glucose deprivation. (**A**) Total protein lactylation levels in BV2 cells during OGD, analyzed by Western blotting (*n* = 3). (**B**) Lactylation at H3K18, H3K56, H3K9, H3K27, H4K16, and H4K8 sites following OGD, shown by Western blotting (*n* = 3). (**C**) Significant increase in nuclear H3K9 lactylation in BV2 cells post-OGD, shown by immunofluorescence staining. Scale bar = 25 μm (*n* = 50). (**D**) H3K9 lactylation in Iba1-labeled microglia from sham-operated and HIE brain tissues, shown by immunofluorescence. Scale bar = 100 μm (*n* = 9). (**E**) Total protein acetylation and H3K9 acetylation levels in BV2 cells post-OGD, analyzed by Western blotting (*n* = 3). Significance: * *p* < 0.05; ** *p* < 0.01; *** *p* < 0.001.

**Figure 4 biomedicines-12-02371-f004:**
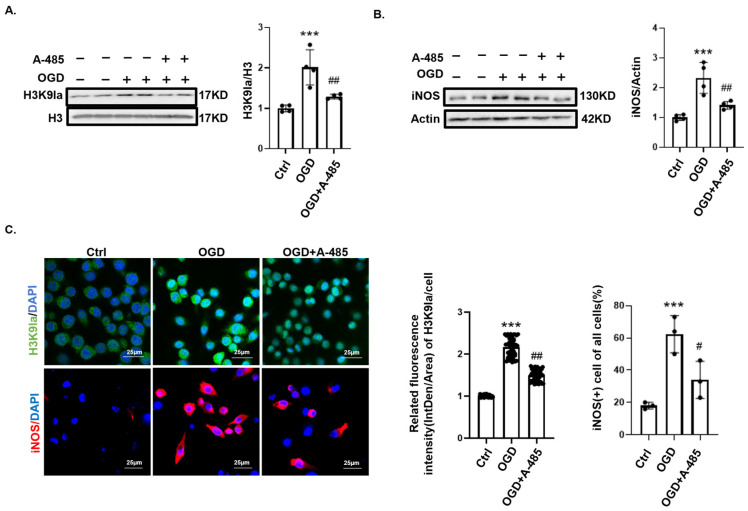
Histone lactylation regulates M1 polarization of microglia. (**A**) H3K9 lactylation levels in BV2 cells treated with or without 50 μM A-485 after 24 h of OGD (*n* = 4). (**B**) Western blotting analysis showed that A-485 reduced M1 polarization levels in cells following OGD (*n* = 4). (**C**) Immunofluorescence indicating that A-485 decreased OGD-induced H3K9 lactylation and reduced iNOS (+) microglia. Scale bar = 25 μm (*n* = 50). Significance: ^#^ *p* < 0.05; ^##^ *p* < 0.01; *** *p* < 0.001.

**Figure 5 biomedicines-12-02371-f005:**
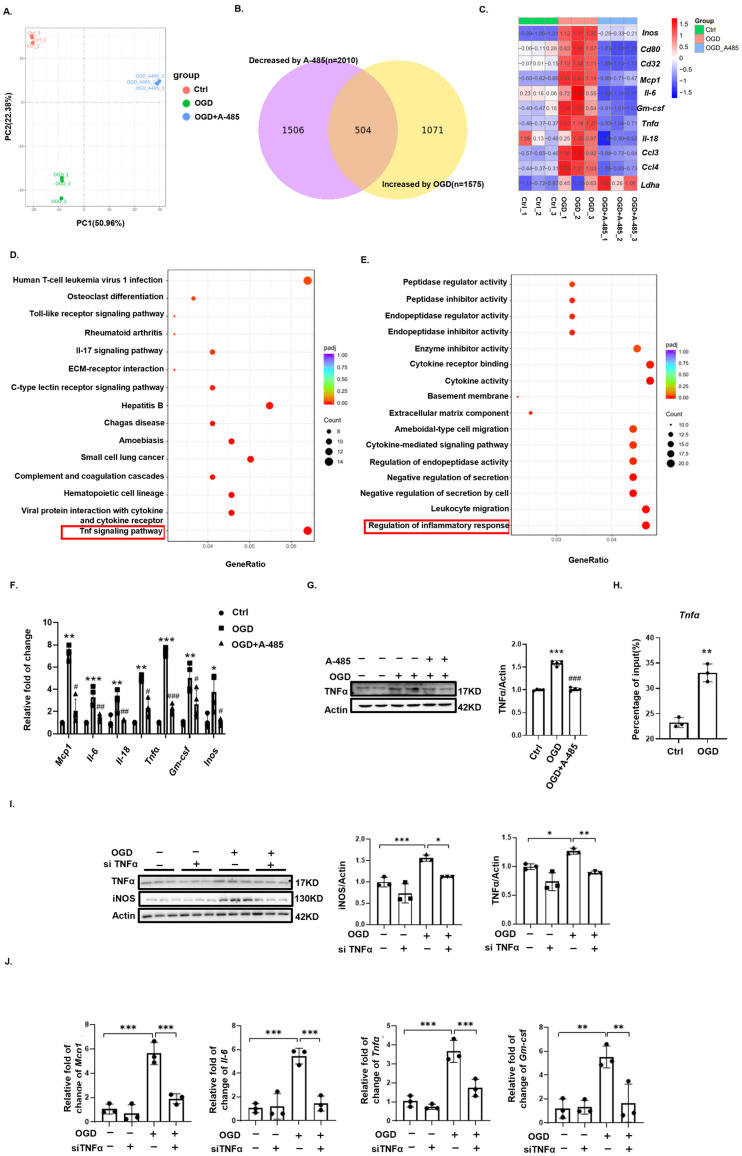
Histone lactylation regulates microglial polarization through the TNFα pathway. (**A**) PCA plot of genes from the control, OGD, and OGD + A-485 groups. (**B**) Venn diagram of genes upregulated in the OGD group compared to the control group and downregulated in the OGD + A-485 group compared to the OGD group. (**C**) M1 microglia-specific genes among the control, OGD, and OGD + A-485 groups. (**D**) KEGG enrichment analysis of intersecting genes from the Venn diagram. (**E**) GO enrichment analysis of intersecting genes from the Venn diagram. (**F**) Western blotting analysis of TNFα levels in the control, OGD, and OGD + A-485 groups (*n* = 4). (**G**) Changes in inflammatory factors among the control, OGD, and OGD + A-485 groups (*n* = 4). *, **, ***, OGD vs. Ctrl; #, ##, ###, OGD + A-485 vs. OGD. (**H**) ChIP-qPCR analysis of H3K9 lactylation enrichment in the *Tnfα* DNA fragment (*n* = 3). (**I**) Changes in M1 microglial polarization among the control, siTNFα, OGD, and OGD + siTNFα groups (*n* = 3). (**J**) Changes in inflammatory factors among the control, siTNFα, OGD, and OGD + siTNFα groups (*n* = 3). Significance: * *p* < 0.05, ** *p* < 0.01, *** *p* < 0.001; # *p* < 0.05, ## *p* < 0.01, ### *p* < 0.001.

**Figure 6 biomedicines-12-02371-f006:**
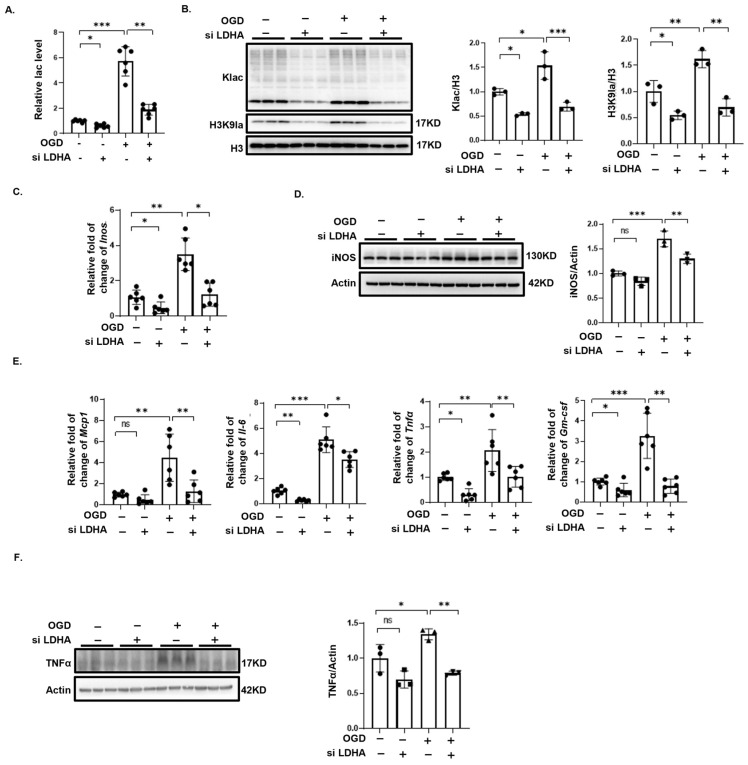
siRNA knockdown of LDHA alleviates OGD-induced M1 microglial polarization and inflammation. (**A**) LDHA knockdown significantly reduces lactate levels in BV2 cells post-OGD (*n* = 6). (**B**) Western blotting showing total protein lactylation and H3K9 lactylation levels in BV2 cells after LDHA knockdown (*n* = 3). (**C**) qPCR results showing reduced OGD-induced expression of M1 microglial markers *Inos* after LDHA knockdown (*n* = 6). (**D**) Western blotting showing reduced iNOS expression LDHA knockdown (*n* = 3). (**E**) qPCR results indicating reduced mRNA levels of pro-inflammatory genes after LDHA knockdown (*n* = 6). (**F**) LDHA knockdown significantly reduced the OGD-induced increase in TNFα (*n* = 3). Significance: * *p* < 0.05; ** *p* < 0.01; *** *p* < 0.001. ns, no significant difference.

## Data Availability

The original data presented in the study are openly available in GEO at GSE272754.

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
