# Peer review of "Oxygen Glucose Deprivation-Induced Lactylation of H3K9 Contributes to M1 Polarization and Inflammation of Microglia Through TNF Pathway"

_biomedicines, 2024, doi:10.3390/biomedicines12102371_

Round 1

Reviewer 1 Report

Comments and Suggestions for Authors

The article is original and very relevant for the field. The authors investigated the role of histone lactylation in hypoxia-induced M1 microglial polarization and inflammation, aiming to provide insights for hypoxic-ischemic encephalopathy (HIE) treatment.

The methodology is very modern, authors using immunohistochemistry, cell biology, modern molecular biology and statistics.

The results showed that, H3K9 lactylation regulated M1 polarization via the TNF signaling pathway. ChIP-qPCR confirmed H3K9 lactylation enrichment at the TNFα locus, promoting OGD-induced M1 polarization and inflammation. These findings may open new therapeutic strategy, more effective and with fewer adverse effects for hypoxic-ischemic encephalopathy of children.

The conclusions are consistent with the evidence and arguments presented.

The references are appropriate, including some relevant authors experience in the field.

I recommend some minor corrections.

1.     In Materials and methods authors musr specify the number of animals used per group. Etical approval should be also mentioned

References- should be written following carefully the Guide for authors for MDPI journals. Many references are mentioning just first 3 authors et al. Just if a paper has more than 10 authors, they are ,mentioned as et al.

Comments on the Quality of English Language

 in text references should not be put as superscript

Author Response

Dear Reviewer:

Thank you for your letter and for the reviewers' comments concerning our manuscript entitled OGD-Induced Lactylation of H3K9 Contributes to M1 Polarization and Inflammation of Microglia Through TNF Pathway (biomedicines-3167204B). Those comments are all valuable and very helpful for revising and improving our paper, as well as the important guiding significance to our research. We have studied comments carefully and have made correction which we hope meet with approval.

For the point-to-point answers, please see the blow content.

  1. In Materials and methods authors must specify the number of animals used per group. Ethical approval should be also mentioned

RE: Thank you for your advice. We have revised the materials and methods. The number of animals used per group have been added in line 79-80, the ethical approval has been added in line 81-82. And the revised content is as follows: “There were 20 rats in sham control group or HIE model group”, “The registration number of the ethical approval is [2023] IACUC Number:3908”.

  1. References-should be written following carefully the Guide for authors for MDPI journals. Many references are mentioning just first 3 authors et al. Just if a paper has more than 10 authors, they are, mentioned as et al.

RE: Thank you for your advice. We are sorry for the mistake. We have revised references in line 442-556.

Reviewer 2 Report

Comments and Suggestions for Authors

He et al. studied the effect of H3K9 lactylation in microglia on their M1 polarization.  It appears that the biochemical data are largely supportive of the thesis, but the morphological data are poor in quality and thus, unconvincing.  However, even the biochemical data suffer from lack of detailed information on how samples were obtained and analyzed; for example, each experimental result shown must be provided with the lengths of treatments (GOD or HIE procedure), how many repeats were performed (n=xx), and how data were normalized.  Without these critical details, it is not possible to evaluate the results (and therefore, the study as a whole).  There are also serious conceptual, logistic and methodological problems, which are described below.

1.     Abstract, lines 22-24. “Using Sprague-Dawley rats to model neonatal HIE, we assessed oxygen glucose deprivation (OGD)'s effects on microglial polarization.”  Besides this sentence being confusing (therefore, should be re-written and clearly state that OGD experiments were done on cultured microglial cells), the author seem to suggest that the OGD-treated microglia culture is a model for HIE.  This assumption is conceptually incorrect and cannot be accepted.  HIE is a whole tissue response in which several other cell types (such as neuronal cells, vessel cells, etc.), in addition to microglia, are present.  What were detected biochemically in microglial cells in vitro could also occur in other cell types, and together, contributes to HIE in vivo.  Unless the biochemical response to OGD is unique to BV2 cells and microglia (i.e. other cell types do not respond in the same manner), any comments/suggestion that microglia H3K9 lactylation is the cause of HIE must be deleted throughout the paper.

2.     Line 45.  It would be good to also briefly explain what M2 is, since works by Mao and Zhao are discussed later.

3.     Materials and Methods.  Lactate measurement is a key method.  Describe how the samples were prepared, how data from different samples were normalized, ect.

4.     Materials and Methods.  For all the antibodies used, give the animals in which they are made, and show that animal species of the primary and secondary antibodies match.

5.     Materials and Methods.  When siRNA and inhibitors are used to treat cells, describe cell viability at the time of sample collection.  If there is a significant cell death, data from such experiments should be treated with caution.  Provide such information in the text.

6.     Line 187.  “HIE Promotes M1 Polarization of Microglia.”  Conceptually, this is opposite of the statement made in the first sentence in the abstract, “ Hypoxia-induced M1 polarization of microglia and resultant inflammation are key mechanisms in hypoxic-ischemic encephalopathy (HIE).”  Which idea is correct?  However, for the reason given in Comment #1, microglia polarization is probably not the only cause.

7.     Line 188.  “To determine whether microglia undergo polarization and identify the type in HIE…”  What are the types of HIE?  This is a new concept that needs explanation.

8.     Fig. 1.  Micrographs are of poor quality, especially A and D.  The quantification in B: it is not clear what the y-axis is.  “Related fluorescence…”?  What is this?  No scale bars are provided.  Presumably, C shows confocal images.  There are some issues.  First, are they through-focused images or single optical sections?  Either way, the quality is super poor.  Second, CD86 should be concentrated on the plasma membrane, but the staining is cytoplasmic and there is no concentration at the cell boarder.  Explain the staining pattern.  I seriously doubt the staining being specific.  For all experiments, give what n is.

9.     Fig. 2.  A-D: Normalization method?  n?  F: The HIE procedure seems to increase the number of microglial cells.  Is this true?  What is the Pearson coefficient for colocalization?  According to these images, LDHA comes from non-microglial cells.  Wouldn’t this change your concept and interpretation regarding the closeness of in vitro and in vivo models?  No scale bars.

10. I am not going to describe individual issues on presentation of data in the rest of Figures, but all the similar comments made for Figs. 1 and 2 apply in general.  Please modify accordingly.  Again, all the morphological data are of poor quality and any measurements made on poor images are not reliable.  Provide better images for all micrographs.

Comments on the Quality of English Language

journal English editor should go over the MS.

Author Response

Dear Reviewer:

Thank you for your letter and for the reviewers' comments concerning our manuscript entitled OGD-Induced Lactylation of H3K9 Contributes to M1 Polarization and Inflammation of Microglia Through TNF Pathway (biomedicines-3167204B). Those comments are all valuable and very helpful for revising and improving our paper, as well as the important guiding significance to our research. We have studied comments carefully and have made correction which we hope meet with approval.

For the point-to-point answers, please see the blow content.

  1. Abstract, lines 22-24. “Using Sprague-Dawley rats to model neonatal HIE, we assessed oxygen glucose deprivation (OGD)'s effects on microglial polarization.” Besides this sentence being confusing (therefore, should be re-written and clearly state that OGD experiments were done on cultured microglial cells), the author seem to suggest that the OGD-treated microglia culture is a model for HIE. This assumption is conceptually incorrect and cannot be accepted.  HIE is a whole tissue response in which several other cell types (such as neuronal cells, vessel cells, etc.), in addition to microglia, are present.  What were detected biochemically in microglial cells in vitro could also occur in other cell types, and together, contributes to HIE in vivo.  Unless the biochemical response to OGD is unique to BV2 cells and microglia (i.e. other cell types do not respond in the same manner), any comments/suggestion that microglia H3K9 lactylation is the cause of HIE must be deleted throughout the paper.

RE: We apologize for the ambiguity caused by our inappropriate description. We have revised the mistake and corrected in line 24-26. The revised manuscript is as follows: In this study, we assessed the effects of hypoxia deprivation on microglial polarization using both an HIE animal model and an oxygen-glucose deprivation cell model. We agree that the OGD model of microglia is not representative of the HIE model, and that the ODG model of microglia simulate only a fraction of the HIE. Apologize again for our inappropriate description.

  1. Line 45. It would be good to also briefly explain what M2 is, since works by Mao and Zhao are discussed later.

RE: Thank you for your advice. We have revised and added a briefly description of M2 microglia in line 51.

  1. Materials and Methods. Lactate measurement is a key method. Describe how the samples were prepared, how data from different samples were normalized, ect.

RE: Thank you for your advice. We have revised and supplemented the lactate measurement in line 112-128.

4.Materials and Methods. For all the antibodies used, give the animals in which they are made, and show that animal species of the primary and secondary antibodies match.

RE: Thank you for your advice. We have revised and added more details of the antibodies in line 145-148,163-165.

  1. Materials and Methods. When siRNA and inhibitors are used to treat cells, describe cell viability at the time of sample collection. If there is a significant cell death, data from such experiments should be treated with caution. Provide such information in the text.

RE: Thank you for your advice. We have revised and added more details of cells' states in line 173-174. The revised manuscript is as follows: After 24-hour transfection, the cells showed normal morphology and no significant cell death was observed, and then cells were treated under normal control condition (DMEM medium, 10% FBS) or OGD condition.

  1. Line 187. “HIE Promotes M1 Polarization of Microglia.” Conceptually, this is opposite of the statement made in the first sentence in the abstract, “ Hypoxia-induced M1 polarization of microglia and resultant inflammation are key mechanisms in hypoxic-ischemic encephalopathy (HIE).”  Which idea is correct?  However, for the reason given in Comment #1, microglia polarization is probably not the only cause.

RE: Thank you for your comment. We apologize for the confusion caused by our inappropriate description. In HIE, there are many factors such as hypoxia, increased inflammatory factors, increased intracellular Ca2+, etc. contribute to M1 polarization of microglial (Mol Neurobiol. 2024 Jul 29., Front Cell Neurosci. 2019 May 24:13:237.), meanwhile, M1 polarization of microglial further aggravate HIE injury by causing neuronal apoptosis, inhibiting neurogenesis and so on (Mol Neurobiol. 2024 Jul 29., Cell Mol Immunol. 2020 Sep;17(9):976-991.). Therefore, M1 polarization of microglial is caused by HIE and in turn promotes the progression of HIE. To avoid confusion, we have revised the manuscript in line 19-20,216. The revised manuscript is as follows:

“Hypoxia-induced M1 polarization of microglia and resultant inflammation take part in the damage caused by hypoxic-ischemic encephalopathy (HIE)” is in line 20-19.

“Hypoxia Promotes M1 Polarization of Microglia” is in line 217

  1. Line 188. “To determine whether microglia undergo polarization and identify the type in HIE…” What are the types of HIE?  This is a new concept that needs explanation.

RE: We apologize for the ambiguity caused by our inappropriate description. The meaning of this statement is to determine the type of microglial polarization in HIE. We have revised the mistake and corrected in line 218-219. The revised manuscript is “To determine whether microglia undergo polarization and identify the type of microglia polarization in HIE”

8.1 Fig. 1.  Micrographs are of poor quality, especially A and D. 

RE: Thank you for your advice. We prepared high resolution figures and replaced them in fig.1B&C,2F,3C&D,4C.

8.2 The quantification in B: it is not clear what the y-axis is.  “Related fluorescence…”?  What is this?  No scale bars are provided.

RE: Thank you for your advice. The y-axis in Fig. 1B is Related fluorescence intensity (IntDen/Area) of CD86 which means that relative fluorescence intensity relative to the mean of the sham group. We have prepared high resolution figures and replaced them in Fig. 1B. The scale bar was in the lower right corner of the picture, and we have replaced it with a more obvious scale bar.

8.3 Presumably, C shows confocal images.  There are some issues.  First, are they through-focused images or single optical sections?  Either way, the quality is super poor.  Second, CD86 should be concentrated on the plasma membrane, but the staining is cytoplasmic and there is no concentration at the cell boarder.  Explain the staining pattern.  I seriously doubt the staining being specific.  For all experiments, give what n is.

RE: Thank you for your advice. We prepared high resolution figures and replaced them in fig.1D. On the question that CD86 is not concentrated on the plasma membrane, it may be due to poor localization of the antibody. In revised manuscript, CD86 has been replaced with iNOS in Fig.1D. And the n of all experiments has been supplemented in figure legends.

9.1 Fig. 2.  A-D: Normalization method?  n?  F: The HIE procedure seems to increase the number of microglial cells.  Is this true?  What is the Pearson coefficient for colocalization? 

RE: Thank you for your advice. The normalization method has been supplemented in line 108,127-128,189. The Images of increased numbers of HIE have also been present in other studies (J Neuroinflammation. 2020 Jun 10;17(1):182., J Neuroinflammation. 2021 Jan 19;18(1):26.). And the statistical analysis of Fig. 2F has been supplemented.

9.2 According to these images, LDHA comes from non-microglial cells. Wouldn’t this change your concept and interpretation regarding the closeness of in vitro and in vivo models?  No scale bars.

RE: Thank you for your comment. We again apologize for the inappropriate description in the abstract. We absolutely agree with you that HIE is a whole-tissue reaction, and several other cell types (e.g., neuronal cells, vascular cells, etc.) are present in addition to microglia. The scale bar was in the lower right corner of the picture, and we have replaced it with a more obvious scale bar.

  1. I am not going to describe individual issues on presentation of data in the rest of Figures, but all the similar comments made for Figs. 1 and 2 apply in general. Please modify accordingly. Again, all the morphological data are of poor quality and any measurements made on poor images are not reliable.  Provide better images for all micrographs.

RE: We have caused you trouble due to low resolution of images. We apologize for that negligence. We have prepared high resolution and replaced all images. We hope for your understanding.

Thank you very much for your attention and time. Look forward to hearing from you.

Yours sincerely

Lu He

Reviewer 3 Report

Comments and Suggestions for Authors

This study shows the microglia in the brain of HIE model rats might polarized to M1 type induced by the histone-lactylattion, which was also verified using BV2 cells. OGD-treated BV2 cells were promoted to M1 polarization through the increase of cytoplasmic lactate by LDHA, H3K9 lactylation and TNF pathway. The story is theoretical according to the results and the methods used are appropriate. I have just a few minor concerns.

1.     Graphic abstract is available for the readers. Might “alleviates” in line 37 be kind of “promotes”?

2.     In the title of graphic abstract, the the differences of the sold arrow and the broken arrow should be explained. 

3.     In the paragraph of “3.2 Lactate Accumulates……”, all the “Fig3”s are Fig2.

4.     In L28, “LDHA” also should be draw with the full name.

5.     In L255, were the other inhibitors (MCE or HY-107455) used in the study? If used, please show the results, too.

6.     The characters in the figures are too small to see.

Author Response

Dear Reviewer:

Thank you for your letter and for the reviewers' comments concerning our manuscript entitled OGD-Induced Lactylation of H3K9 Contributes to M1 Polarization and Inflammation of Microglia Through TNF Pathway (biomedicines-3167204B). Those comments are all valuable and very helpful for revising and improving our paper, as well as the important guiding significance to our research. We have studied comments carefully and have made correction which we hope meet with approval.

For the point-to-point answers, please see the blow content.

  1. Graphic abstract is available for the readers. Might “alleviates” in line 37 be kind of “promotes”?

RE: We apologize for our negligence. We have corrected the mistake. The revised manuscript is “H3K9 lactylation promotes OGD-induced M1 microglial activation and pro-inflammation” in line 41.

  1. In the title of graphic abstract, the differences of the sold arrow and the broken arrow should be explained. 

RE: We apologize for our negligence. The sold arrow means direct effect and the broken arrow means indirect effect. We have revised the graphic abstract and supplemented explanations of the sold arrow and the broken arrow in line 40.

  1. In the paragraph of “3.2 Lactate Accumulates……”, all the “Fig3”s are Fig2.

RE: We apologize for our negligence. We have corrected the mistake in the paragraph of “3.2 Lactate Accumulates……” in line 244-254.

  1. In L28, “LDHA” also should be draw with the full name.

RE: Thank you for your advice. We have revised “LDHA” to “lactate dehydrogenase A” in line 30-31.

  1. In L255, were the other inhibitors (MCE or HY-107455) used in the study? If used, please show the results, too.

RE: We are very sorry that our expression is not clear enough, which has caused your misunderstanding. In this study, we used only A-485 as an inhibitor of P300, and its catalog number is HY-107455, purchased from MCE. We have revised the manuscript in line 282-284. The revised manuscript is as follows: “we used the P300 inhibitor, A-485 (MCE, HY-107455), to significantly suppress OGD-induced H3K9 lactylation”.

  1. The characters in the figures are too small to see.

RE: Thank you for your advice. We have revised and enlarged the characters in the figures.

Thank you very much for your attention and time. Look forward to hearing from you.

Yours sincerely

Lu He

Reviewer 4 Report

Comments and Suggestions for Authors In the current study, a research group from Wuhan and Taiyuan investigated the link between histone lactylation and M1 microglial polarization during oxygen-glucose deprivation (OGD), which leads to ischemic encephalopathy. Using a variety of methods, including the application of small selective inhibitors, siRNA technique and RNA-seq, Lu He and colleagues clearly demonstrated that OGD leads to lactylation of H3K9 with subsequent activation of the TNF-alpha signaling axis and acquisition of a pro-inflammatory phenotype by microglial cells both in vitro and in vivo. This work is characterized by high novelty and originality, it is very interesting and can be published in Biomedicines after a major revision. Dear authors, please respond to the following comments:   Major comments: -- RNA-seq analysis data - (a) Please provide a more detailed description of the RNA-seq methodology performed in this study (section 2.10). The information provided is not sufficient to understand the quality of the transcriptome study performed. (b) Please provide information on the fold change of key genes considered in this article (e.g. CD86, pro-inflammatory genes, LDHA) obtained by RNA-seq (are they consistent with qPCR and Western blotting data?). -- Please discuss why knockdown of LDHA leads to a significant decrease in CD86/actin levels in BV2 cells in the absence of OGD (Fig. 6D; please double-check the p-value in Fig. 6C between OGD(-)/siLDHA(-) and OGD(-)/siLDHA(+) groups - in my opinion the difference is statistically significant (same for Fig. 6E (IL-6, Il-1beta, GM-CSF)). -- Please increase the quality of all fluorescence microscopy images. Unfortunately, when zoomed in, only pixels are visible.   Minor comments: Title - please decipher OGD in title Line 76 and throughout the manuscript - please indicate 2 in O2 and N2 as lower case. Lines 133-135, 137 - as I understand it, Abclonal is the correct name (not Abconal or Abclonal). Please correct. Line 177 - Please specify which method and reagent was used for enzyme digestion. Line 180 - please clarify if you mean TNF-alpha as mRNA or protein? To avoid confusion, please indicate the name of genes (not only TNF-alpha) in italics. Section 2.12 - please specify which algorithm was used to estimate the normality of the distribution of the samples analyzed. Line 185, 211, 212 and throughout the manuscript - please write p in italics in the p-value. Figure 1 - please increase the resolution and size of Figure 1A and increase the font size of the captions in all bar plots. The quality of the bar plots is very poor (especially Figure 1F). Please correct. Fig. 1A - according to the formula given in line 99, the authors did not calculate the volume but the area of infarcts. Please correct or provide the formula for calculating infarct volume. Fig. 1E - please swap the bar graphs for CD86/actin and CD206/actin  for consistency of the data presented (as in Fig. 1C). Line 222 - The word "while" contrasts the first part of the sentence with the second part of the sentence. In this case there is no contrast (in both cases there is an increase in LDHA expression). Please choose a more appropriate word. Figure 5C - The labels above the Venn diagram are poorly understood. Please correct.

Author Response

Dear Reviewer:

Thank you for your letter and for the reviewers' comments concerning our manuscript entitled OGD-Induced Lactylation of H3K9 Contributes to M1 Polarization and Inflammation of Microglia Through TNF Pathway (biomedicines-3167204B). Those comments are all valuable and very helpful for revising and improving our paper, as well as the important guiding significance to our research. We have studied comments carefully and have made correction which we hope meet with approval.

For the point-to-point answers, please see the blow content.

Major comments: -- RNA-seq analysis data –

  • Please provide a more detailed description of the RNA-seq methodology performed in this study (section 2.10). The information provided is not sufficient to understand the quality of the transcriptome study performed.

RE: Thank you for your advice. We have revised the 2.10 RNA Extraction, Library Preparation, and Sequencing. The revised manuscript is as follows:

“According to the manufacturer’s instructions, RNA integrity was assessed using the RNA Nano 6000 Assay Kit of the Bioanalyzer 2100 system (Agilent Technologies, CA, USA). Clean data were obtained by removing reads containing adapter, reads containing N base and low-quality reads from raw data. FeatureCounts v1.5.0-p3 was used to count the reads numbers mapped to each gene. And then FPKM of each gene was calculated based on the length of the gene and reads count mapped to this gene. The R Bioconductor package DESeq2 was utilized to screen out DEGs. The p value <0.01 and |log2fc|>0.5 were set as the cut-off criteria for identifying DEGs. Gene Ontology (GO) terms and Kyoto Encyclopedia of Genes and Genomes (KEGG) pathways were identified using the KOBAS 2.0 server to sort out functional categories of DEGs.” is in line 192-200.

  • Please provide information on the fold change of key genes considered in this article (e.g. CD86, pro-inflammatory genes, LDHA) obtained by RNA-seq (are they consistent with qPCR and Western blotting data?).

RE: We apologize for ignoring this question before. After carefully checking the fold change of key genes obtained by RNA-seq. we found that the expression of most genes remained consistent. We have supplemented the fold change of key genes in Fig. 5C. But we found that marker of M1 microglial polarization Cd86 did not significantly increase in OGD group which was not fully consistent with qPCR and Western blotting data. But, other makers of M1 microglial polarization (Inos, Cd32, Cd80) significantly increase in OGD group. We verified the protein and mRNA expression of Inos, which was most significantly changed in RAN-seq, and found that all matched. There have been many reports on Inos as a marker of M1 polarization of microglia (J Neuroinflammation. 2021 Nov 13;18(1):267, CNS Neurosci Ther. 2023 Dec;29(12):4113-4123.). Therefore, we have replaced the marker of M1 microglial polarization with Inos in revised manuscript. We hope for your understanding.

  • Please discuss why knockdown of LDHA leads to a significant decrease in CD86/actin levels in BV2 cells in the absence of OGD (Fig. 6D; please double-check the p-value in Fig. 6C between OGD(-)/siLDHA(-) and OGD(-)/siLDHA(+) groups - in my opinion the difference is statistically significant (same for Fig. 6E (IL-6, Il-1beta, GM-CSF)).

RE: Thank you for your question. As for the reason why knockdown of LDHA leads to a significant decrease in CD86/actin levels in BV2 cells in the absence of OGD, we think that this is because there is a basal level of H3K9 lactacylation modification in BV2 cells under normal conditions, and knockdown of LDHA under normal conditions also reduced H3K9 lactate level in Fig.6B. Therefore, knockdown of LDHA leads to a significant decrease in CD86/actin levels in the presence or absence of OGD. To double-check the p-value in Fig. 6, we have again prepared cell samples and done experiments (n=3). When the obtained data have been combined with the previous data, there has been statistically significant between OGD(-)/siLDHA(-) and OGD(-)/siLDHA(+) groups in IL-6,TNFα,GM-CSF and level of lactate. The revised manuscript has been replaced in Fig. 6.

  • Please increase the quality of all fluorescence microscopy images. Unfortunately, when zoomed in, only pixels are visible.  

RE: Thank you for your advice. We prepared high resolution figures and replaced them in fig.1B&C,2F,3C&D,4C.

 Minor comments:

Title

- please decipher OGD in title Line 76 and throughout the manuscript

RE: Thank you for your advice. We have corrected the mistake and replaced OGD with oxygen glucose deprivation in line 83.

- please indicate 2 in O2 and N2 as lower case.

RE: Thank you for your advice. We have revised and indicated 2 in O2 and N2 as lower case in line 77,88,90.

Lines 133-135, 137

- as I understand it, Abclonal is the correct name (not Abconal or Abclonal). Please correct.

RE: We apologize for our negligence. We have corrected the mistake in line 153-158,179,202,210.

Line 177

- Please specify which method and reagent was used for enzyme digestion.

RE: We apologize for our negligence. The Chromatin immunoprecipitation was conducted using commercial ChIP kit (Abclonal, RK20258) where is the reagent used for enzyme digestion from. We have revised manuscript and added this information in line202-203.

Line 180

- please clarify if you mean TNF-alpha as mRNA or protein? To avoid confusion, please indicate the name of genes (not only TNF-alpha) in italics.

RE: Thank you for your advice. We have corrected the mistake throughout the manuscript. All names of genes have been italicized. Besides, the first letters have been uppercased and the rest have been lowercased to distinguish between mouse and human genes. All our revised manuscript is highlighted in green.

Section 2.12

- please specify which algorithm was used to estimate the normality of the distribution of the samples analyzed.

RE: Thank you for your advice. We have revised manuscript in line213-214. In the revised manuscript, statistical significance was calculated using a one-way analysis of variance (ANOVA).

 Line 185, 211, 212 and throughout the manuscript

- please write p in italics in the p-value.

RE: We apologize for our negligence. We have corrected the mistake throughout the manuscript. All our revised manuscript is highlighted in green.

Figure 1

- please increase the resolution and size of Figure 1A and increase the font size of the captions in all bar plots. The quality of the bar plots is very poor (especially Figure 1F). Please correct.

RE: Thank you for your advice. We prepared high resolution figures and replaced them in fig1A&B&C,2F,3C&D,4C. And we enlarged the characters in the figures.

Fig. 1A

- according to the formula given in line 99, the authors did not calculate the volume but the area of infarcts. Please correct or provide the formula for calculating infarct volume.

RE: We apologize for our negligence. In this study, we quantified OGD-mediated brain damage by calculating the area of infarcts. However, we made a mistake in Fig 1A. Therefore, it’s correct in line 99. We have revised the statistical plot in Fig.1A.

Fig. 1E

- please swap the bar graphs for CD86/actin and CD206/actin for consistency of the data presented (as in Fig. 1C).

RE: Thank you for your advice. We have swapped he bar graphs for CD86/actin and CD206/actin in Fig.1E.

Line 222

– The word "while" contrasts the first part of the sentence with the second part of the sentence. In this case there is no contrast (in both cases there is an increase in LDHA expression). Please choose a more appropriate word.

RE: Thank you for your advice. We have revised the word "while" to “and” in line 251.

Figure 5C

- The labels above the Venn diagram are poorly understood. Please correct.

RE: Thank you for your advice. We have revised the labels above the Venn diagram in Figure.5B, to make it clearer.

Thank you very much for your attention and time. Look forward to hearing from you.

Yours sincerely

Lu He

Round 2

Reviewer 2 Report

Comments and Suggestions for Authors

Although the authors have made certain changes, the extent of modifications is insufficient.  For example, in my earlier review, I asked authors to provide much more precise information on each experiment: the time of sample collection (such as x hours of OGD, etc.), the concentrations and lengths of drug treatment (such as x mM of A-485 for y hours, concentrations of siRNAs, etc.) and other precise experimental conditions.  However they were not provided either in figure legends (or in the text) although authors did provide the sample number information.  These pieces of information are critical for at least two reasons.  First, the results shown in this paper can be repeated by other investigators (is they want to do so) to confirm your observation, so it is your responsibility to describe precisely how all the experiments were performed.  Second, readers need to know how experiments were done so that they can interpret the results.  Without knowing how the data were obtained, one cannot interpret them.  Once again, please provide detailed information regarding the way samples were prepared for all the results shown in figures.  As you will see below, there are other issues that need to be corrected.

1.     Line 25.  What is “hypoxia deprivation”?  Depriving hypoxia means to me hyperoxia (high concentration of oxygen).  Is this paper on the effects of hyperoxia?

2.     Figure 1.  I see that the brain depicting hypoxic-ischemic encephalopathy and the M1 microglia come from papers published by other investigators: the brain from a paper by Dordoe et al. (https://doi.org/10.1016/j.cytogfr.2023.07.005) and the cell from a paper by Haupt et al. (The dual role of microglia in ischemic stroke and its modulation via extracellular vesicles and stem cells).  Have the authors obtained permission to use these figures?  If they were copied without permissions, the act can be regarded as plagiarism.   How about other images?  If they were also copied, the source must be identified and proper permissions must be obtained.  The best is to make your own graphics.

3.     Line 63.  “During HIE, hypoxia induces significant lactate accumulation in microglia).  This sentence needs a reference.

4.     Line 116.  “Frozen hippocampal tissue was grinded at 2500rpm for 10 min and…”  How much (in mass) hippocampal tissue was used?  Why hippocampal tissue?  Explain.  What instrument was used?  A tissue blender?

5.     Line 123.  “the optical density (OD)…”  Change this to “absorbance”.  Also, change all OD to absorbance.

6.     Lines 126-127.  What is standard?  Describe and define.

7.     Immunofluorescence Microscopy.  For every confocal image, describe whether it is a single optical section or a through-focus (i.e. stacked) image.  For fluorescence quantification, errors may arise if a single optical section is used.

8.     Line 222.  Is iNOS a surface marker?  This is news to me.  This expression appears in other places throughout the paper.  If it is a surface marker, the Fig. 1D OGD panel fails to support this statement!

9.     Fig. 2.  As I stated at the beginning, each data must be accompanied by how samples were prepared.  As I pointed out in my first review, Panel F seems to show HIE procedure increased microglia numbers.  Where do they come from?  The HIE sample does not show good confocal effects.  Replace with a better image? 

10. Lines 251-252.  Authors state, “immunofluorescence indicated a marked increase in LDHA in Iba1- positive microglia within the HIE animal model.”  I am not convinced of this observation because where there is red, I see rather few yellow dots (i.e. colocalization).  Moe convincing demonstration is needed.  Pearson coefficient requires more discrete localization images.  The HIE image is too diffuse (unfocused?) for this analysis.  What do the 4 samll images show?  Explain.

11. Fig. 3D.  There is a lot of green without red in HIE, indicating that H3K9 lactylation occurs more in non-microglial cells.  This suggests microglia may not be the main player of neuroinflammation.  What are these cells?  They seem to express high levels of H3K9A.  This image seems to argue against the big theme that microglia via H3K9 lactylation play a major role of neuroinflammation.  Both this figure and Fig. 2F seem to downplay the importance of microglia from the mechanistic point of view.

12. Lines 283-284.  This is the only place where authors mention the use of A-485, but they fail to describe how cells were treated by the inhibitor.  Describe how the inhibitor experiments were done here or in Materials and Methods.  Give concentrations, durations, timing relative to OGD treatment, etc.

13. Fig. 4B.  This result is not convincing.

14. Line 296.  Change “differentially expressed genes” to “differentially expressed genes (DEGs).

15. Line 298.  “identifying 504 genes through Venn diagram analysis (Fig. 5B).”  Venn diagram is not an analytical method.  It is a graphic method to show the results of comparative analyses.  Rewrite.

16. Fig. 5G.  Looking at the gels and the graph, I am not convinced of the quantification and also such high significance values.

17. For all graphs, it would help greatly if individual data points are superimposed as dots.  This may resolve some questionable data.

18. Capitalization of words in titles is inconsistent.

Comments on the Quality of English Language

A careful editing by someone who understands the content is required.   There are cases where wrong words and terms are used some, and tenses must be checked.  Grammar  must be also checked.

Author Response

Dear Reviewer:

Thank you for your letter and comments concerning our manuscript entitled OGD-Induced Lactylation of H3K9 Contributes to M1 Polarization and Inflammation of Microglia through TNF Pathway (biomedicines-3167204B). According to your advices, we have further improved the method and the text in revised manuscript. Additionally, with the assistance of professional English editors, we have made language revisions to the manuscript.

For the point-to-point answers, please see the blow content.

  1. Line 25.  What is “hypoxia deprivation”?  Depriving hypoxia means to me hyperoxia (high concentration of oxygen).  Is this paper on the effects of hyperoxia?

RE: We apologize for our negligence. We have corrected the mistake and revised in line 25.

  1. Figure 1.  I see that the brain depicting hypoxic-ischemic encephalopathy and the M1 microglia come from papers published by other investigators: the brain from a paper by Dordoe et al. (https://doi.org/10.1016/j.cytogfr.2023.07.005) and the cell from a paper by Haupt et al. (The dual role of microglia in ischemic stroke and its modulation via extracellular vesicles and stem cells).  Have the authors obtained permission to use these figures?  If they were copied without permissions, the act can be regarded as plagiarism.   How about other images?  If they were also copied, the source must be identified and proper permissions must be obtained.  The best is to make your own graphics.

RE: Thank you for your comment. Figure 1 is our experimental data. In my understanding, the graphic you refer is graphical abstract. It was made by ourselves on the website: https://www.biorender.com/academic-license. We have supplemented the agreement number: IF27AZ3WAZ in line 41.

  1. Line 63.  “During HIE, hypoxia induces significant lactate accumulation in microglia).  This sentence needs a reference.

RE: Thank you for your advice. We have added a reference (Biochem Genet. 2024 Jan 21.) and revised in line 62.

  1. Line 116.  “Frozen hippocampal tissue was grinded at 2500rpm for 10 min and…”  How much (in mass) hippocampal tissue was used?  Why hippocampal tissue?  Explain.  What instrument was used?  A tissue blender?

RE: Thank you for your advice. The hippocampal tissue which we used was 10mg. And to ensure that the location of lactate measurement was consistent with the location of immunofluorescence, we chose to sample the hippocampal tissue that is critical for cognitive function. The instrument which we used was high-speed tissue grinder from Servicebio (Wuhan, China) and the product model was KZ-11. We have supplemented the mass of hippocampal tissue and the information of high-speed tissue grinder in lines 115-116.

  1. Line 123.  “the optical density (OD)…”  Change this to “absorbance”.  Also, change all OD to absorbance.

RE: Thank you for your advice. We have changed all OD to absorbance, and revised in lines 123-127.

  1. Lines 126-127.  What is standard?  Describe and define.

RE: Thank you for your advice. The standard was 3mM lactic acid which came from L-lactate assay kit (Cat# A019-2-1, Nanjing Jiancheng Bioengineering Institute, China). We have revised and supplemented the description in lines 120-121.

  1. Immunofluorescence Microscopy.  For every confocal image, describe whether it is a single optical section or a through-focus (i.e. stacked) image.  For fluorescence quantification, errors may arise if a single optical section is used.

RE: Thank you for your advice. We have added a description about the through-focus (i.e. stacked) image in Materials and Methods, and revised in lines (145-146).

  1. Line 222.  Is iNOS a surface marker?  This is news to me.  This expression appears in other places throughout the paper.  If it is a surface marker, the Fig. 1D OGD panel fails to support this statement!

RE: We apologize for our negligence. iNOS is a marker of M1 microglia but not a surface maker. We have corrected this mistake and revised in lines 235,298.

  1. 2.  As I stated at the beginning, each data must be accompanied by how samples were prepared.  As I pointed out in my first review, Panel F seems to show HIE procedure increased microglia numbers.  Where do they come from?  The HIE sample does not show good confocal effects.  Replace with a better image? 

RE: Thank you for your advice. We have supplemented more details about how samples were prepared in Materials and Methods, the revised manuscript is in lines 101-103,111,115-116,131,155-158,184-187,215-216. About panel F, we have re-prepared two images that had similar numbers of microglia.

  1. Lines 251-252.  Authors state, “immunofluorescence indicated a marked increase in LDHA in Iba1- positive microglia within the HIE animal model.”  I am not convinced of this observation because where there is red, I see rather few yellow dots (i.e. colocalization).  Moe convincing demonstration is needed.  Pearson coefficient requires more discrete localization images.  The HIE image is too diffuse (unfocused?) for this analysis.  What do the 4 samll images show?  Explain.

RE: Thank you for your advice. We have replaced Fig. 2F with a more convincing image. And we have added an explanation for the 4 small images in Fig. 2F.

  1. 3D.  There is a lot of green without red in HIE, indicating that H3K9 lactylation occurs more in non-microglial cells.  This suggests microglia may not be the main player of neuroinflammation.  What are these cells?  They seem to express high levels of H3K9A.  This image seems to argue against the big theme that microglia via H3K9 lactylation play a major role of neuroinflammation.  Both this figure and Fig. 2F seem to downplay the importance of microglia from the mechanistic point of view.

RE: Thank you for your comments. Based on your comments, we have changed the images and adjusted the contrast to reduce the lower H3k9 lactate signal. In our study, increased H3K9 lactylation in microglia could promote neuroinflammation. However, whether H3K9 lactylation in other cells is associated with neuroinflammation is unknown. Of course, we do not deny that other cells may be involved in neuroinflammation either through or independent of H3K9 lactylation.

  1. Lines 283-284.  This is the only place where authors mention the use of A-485, but they fail to describe how cells were treated by the inhibitor.  Describe how the inhibitor experiments were done here or in Materials and Methods.  Give concentrations, durations, timing relative to OGD treatment, etc.

RE: Thank you for your advice. We have supplemented the concentrations, durations, timing of A-485, and revised in 295-296.

  1. 4B.  This result is not convincing.

RE: Thank you for your advice. We have replaced a more convincing result                     in Fig. 4B.

  1. Line 296.  Change “differentially expressed genes” to “differentially expressed genes (DEGs).

RE: Thank you for your advice. We have changed “differentially expressed genes” to “differentially expressed genes (DEGs).” in line 308

  1. Line 298.  “identifying 504 genes through Venn diagram analysis (Fig. 5B).”  Venn diagram is not an analytical method.  It is a graphic method to show the results of comparative analyses.  Rewrite.

RE: We apologize for our negligence. We have corrected this mistake and revise in line 310. The revised manuscript is “identifying 504 genes and showing in Venn diagram”.

  1. 5G.  Looking at the gels and the graph, I am not convinced of the quantification and also such high significance values.

RE: Thank you for your comment. We have double checked the statistical results of Fig. 5G and corrected in revised manuscript.

  1. For all graphs, it would help greatly if individual data points are superimposed as dots.  This may resolve some questionable data.

 RE: Thank you for your advice. We have replaced all graphs and the individual data points have been superimposed as dots in revised manuscript.

  1. Capitalization of words in titles is inconsistent.

RE: We apologize for our negligence. We have corrected those mistakes and revised in lines 71,149.

We are truly grateful for your valuable assistance and suggestions on our manuscript, which have been immensely meaningful to us. Wishing you continued success in your work.

Reviewer 4 Report

Comments and Suggestions for Authors

I express my respect and appreciation to the authors for their careful attention to my comments. All the major shortcomings I pointed out have been fully corrected. The article is ready for publication after correction of two minor revisions:

1. Fig 1 (line 234), Fig 2 (line 256), Fig 4 (line 290), Fig 5 (line 320), Fig 6 (line 345). Please write these names correctly with dot and space as Fig 1, Fig 2, Fig 3, etc.

2. All figure legends. Please add a space before (n = X) (e.g., "um (n = 3)" (line 237), "tissues (n = 3)" (line 238), "OGD (n = 3)" (line 241), etc.). 

I wish the authors good luck in their further research! 

Author Response

Dear Reviewer:

Thank you for your letter and comments concerning our manuscript entitled OGD-Induced Lactylation of H3K9 Contributes to M1 Polarization and Inflammation of Microglia Through TNF Pathway (biomedicines-3167204B).

For the point-to-point answers, please see the blow content.

  1. Fig 1 (line 234), Fig 2 (line 256), Fig 4 (line 290), Fig 5 (line 320), Fig 6 (line 345). Please write these names correctly with dot and space as Fig 1, Fig 2, Fig 3, etc.

RE: We apologize for our negligence. We have corrected those mistakes and revised in lines 246,268,284,302,332,357.

  1. All figure legends. Please add a space before (n = X) (e.g., "um (n = 3)" (line 237), "tissues (n = 3)" (line 238), "OGD (n = 3)" (line 241), etc.). 

 RE: Thank you for your advice. We have added a space before (n=X) and revised all figure legends.

Appreciate for your help in our research, which means a lot to us. We wish you all the best in your work and happiness in your life.

Round 3

Reviewer 2 Report

Comments and Suggestions for Authors

The authors made efforts to revise the manuscript, to which this reviewer would like to commend.  Unfortunately, however, there are new issues caused by perhaps during the process of revision (for detail, see below).  When comparing the same graphs in the original and current manuscripts, there are many small, but clear differences in values depicted in various graphs.  Which graph of the same data is correct?  This reviewer feels that this is evidence for data reliability being compromised.  To simply put, I can no longer trust what is shown as data, especially data analyses by statistical means.  In addition, as a scientific document, the paper still has language issues.

1.        My lack of confidence in the study originates from authors’ response to my following question: “Looking at the gels and the graph in Fig. 5G, I am not convinced of the quantification and also such high significance values.”  Authors responded, “Thank you for your comment. We have double checked the statistical results of Fig. 5G and corrected in revised manuscript.”  This sounds very innocent and simple as they have corrected Fig. 5G, which now looks very different.  From my long experience as a reviewer, a typical authors’ response to my comment would be to first describe logically why the mistake had happened and then describe how the mistake was corrected.  Such a response gives a reviewer confidence that the mistake was properly corrected.  However, because these authors did not follow this pattern of response, a reviewer has no idea if the mistake was properly corrected.  Besides, the authors should have caught the discrepancy between the gel data and the graph before submitting the paper.  This episode made me go through quickly all the graphs and see if the graphs in the revised manuscript and those in the original submission were the same.  Very surprisingly, I found many mismatches in bar lengths (i.e. measured values) although they are not huge but the differences are clearly detectable.  Such alterations occurred in the following graphs:

i.       Fig. 1 A, B, C (right), and E (left)

ii.      Fig. 2B

iii.      Fig. 4C (Y-axis scale is altered)

iv.      Fig. 5G (already discussed)

v.      Fig. 6A (Y-axis scale changed)

2.       These changes were not listed as alterations in the revised version.  How did they occur?  Why were these changes made?  Since these alterations were made without mentioning, it made me think that the data presented may not be fully reliable including all the statistical analyses performed.  Although it is hardly discussed, science depends on investigator’s correct presentation and reliable analyses of data.   I am sorry, but this rule may have been violated.

3.       The morphological data now look better, but then I wonder why the authors did not show these data to start with?  There must be reasons for using (to me) less convincing images.  Why?  What were the reasons?

4.       In their response, authors have ignored certain queries.  Why?  They have no answers, or did they think that the questions were not worth responding?  For example, I asked why some BV2 cells did not stain for the M1 marker and provided one possible answer, but authors did not respond to this question.

Comments on the Quality of English Language

As a scientific document, the Materials and Methods section and the Result section do not meet the standard level of writing although Abstract, Introduction and Discussion are fine. 

Author Response

Dear review:

Thank you very much for your advices and help on our manuscript, whether we can eventually publish it on biomedicines or not. Wish you all the best with your work.

The point-by-point responses are shown below

  1. My lack of confidence in the study originates from authors’ response to my following question: “Looking at the gels and the graph in Fig. 5G, I am not convinced of the quantification and also such high significance values.”  Authors responded, “Thank you for your comment. We have double checked the statistical results of Fig. 5G and corrected in revised manuscript.”  This sounds very innocent and simple as they have corrected Fig. 5G, which now looks very different.  From my long experience as a reviewer, a typical authors’ response to my comment would be to first describe logically why the mistake had happened and then describe how the mistake was corrected.  Such a response gives a reviewer confidence that the mistake was properly corrected.  However, because these authors did not follow this pattern of response, a reviewer has no idea if the mistake was properly corrected.  Besides, the authors should have caught the discrepancy between the gel data and the graph before submitting the paper.  This episode made me go through quickly all the graphs and see if the graphs in the revised manuscript and those in the original submission were the same.  Very surprisingly, I found many mismatches in bar lengths (i.e. measured values) although they are not huge but the differences are clearly detectable.  Such alterations occurred in the following graphs:
  2. Fig. 1 A, B, C (right), and E (left)
  3. Fig. 2B

iii.      Fig. 4C (Y-axis scale is altered)

  1. Fig. 5G (already discussed)
  2. Fig. 6A (Y-axis scale changed)

Re: Regarding the concerns raised by the reviewer about the modifications to the graphs, it appears that the issues stem from statistical errors. After a thorough review, we did indeed find errors in the statistical data of Fig. 5G, as pointed out by the reviewer. However, we did not encounter similar problems in our images, and there are significant differences between the three groups.

To prevent further issues, we meticulously reviewed the statistical graphs throughout the manuscript and made the necessary corrections. Specifically, for Fig. 1 (A, B, C - right) and Fig. 1E (left), we have updated the images and revised the results based on the comments provided by other reviewers.  Additionally, we acknowledge that similar statistical errors were found upon review in Fig. 2B, Fig. 4C, and Fig. 6A, akin to those identified in Fig. 5G.  These errors have now been corrected with the utmost care

We sincerely hope there is no misunderstanding and appreciate your attention to these matters.

  1. These changes w ere not listed as alterations in the revised version.  How did they occur?  Why were these changes made?  Since these alterations were made without mentioning, it made me think that the data presented may not be fully reliable including all the statistical analyses performed.  Although it is hardly discussed, science depends on investigator’s correct presentation and reliable analyses of data.   I am sorry, but this rule may have been violated.

Re: Most of these changes have already been made in response to answer other reviewers. We do not believe that the statistical errors have impacted the reliability of our results, as there are significant differences in the Western blotting and fluorescent images. However, we would like to acknowledge that the first author wrote the manuscript formally for the first time, so many problems arose during the process. We deeply regret the lack of rigor at that stage, but we have made continuous improvements throughout the revision process.

We sincerely hope to gain your understanding and appreciate your patience as we work through these issues.

  1. The morphological data now look better, but then I wonder why the authors did not show these data to start with?  There must be reasons for using (to me) less convincing images.  Why?  What were the reasons?

Re: First and foremost, we sincerely apologize for the suboptimal quality of the morphological data initially provided. The low-resolution images submitted in the first version were due to submitted compressed files, a point noted by other reviewers as well. After we re-submitted high-resolution images, this issue was resolved.

We appreciate your understanding and the valuable feedback that has helped us enhance the quality of our work. Additionally, inappropriate contrast led to the presence of numerous non-specific signals in Figures 2F and 3D. Although the co-localization signals in the HIE group were indeed increased in the previous images, the excessive non-specific signals may cause confusion for readers, which we overlooked. We sincerely appreciate your bringing this issue to our attention.

We appreciate your understanding and the valuable feedback that has helped us enhance the quality of our work.

  1. In their response, authors have ignored certain queries.  Why?  They have no answers, or did they think that the questions were not worth responding?  For example, I asked why some BV2 cells did not stain for the M1 marker and provided one possible answer, but authors did not respond to this question.

Re: We read your previous comments carefully again and found no questions about “why some BV2 cells did not stain for the M1 marker”. You pointed it out in the last round of comments that “Is iNOS a surface marker?”. We have revised the wording in round 2. We have carefully addressed and responded to each of your comments and sincerely hope for your understanding and support.